# PixelNN: Example-based Image Synthesis

**Aayush Bansal**    **Yaser Sheikh**    **Deva Ramanan**
Carnegie Mellon University
{aayushb,yaser,deva}@cs.cmu.edu

## Abstract

We present a simple nearest-neighbor (NN) approach that synthesizes high-frequency photorealistic images from an "incomplete" signal such as a low-resolution image, a surface normal map, or edges. Current state-of-the-art deep generative models designed for such conditional image synthesis lack two important things: (1) they are unable to generate a large set of diverse outputs, due to the mode collapse problem. (2) they are not interpretable, making it difficult to control the synthesized output. We demonstrate that NN approaches potentially address such limitations, but suffer in accuracy on small datasets. We design a simple pipeline that combines the best of both worlds: the first stage uses a convolutional neural network (CNN) to map the input to a (overly-smoothed) image, and the second stage uses a pixel-wise nearest neighbor method to map the smoothed output to multiple high-quality, high-frequency outputs in a controllable manner. Importantly, pixel-wise matching allows our method to compose novel high-frequency content by cutting-and-pasting pixels from different training exemplars. We demonstrate our approach for various input modalities, and for various domains ranging from human faces, pets, shoes, and handbags.

## 1 Introduction

We consider the task of generating high-resolution photo-realistic images from *incomplete* input such as a low-resolution image, sketches, surface normal map, or label mask. Such a task has a number of practical applications such as upsampling/colorizing legacy footage, texture synthesis for graphics applications, and semantic image understanding for vision through analysis-by-synthesis. These problems share a common underlying structure: a human/machine is given a signal that is missing considerable details, and the task is to reconstruct plausible details.

Consider the edge map of cat in Figure 1-c. When we humans look at this edge map, we can easily imagine multiple variations of whiskers, eyes, and stripes that could be viable and pleasing to the eye. Indeed, the task of image synthesis has been well explored, not just for its practical applications but also for its aesthetic appeal.

**GANs:** Current state-of-the-art approaches rely on generative adversarial networks (GANs) (Goodfellow et al., 2014), and most relevant to us, *conditional* GANS that generate image conditioned on an input signal (Denton et al., 2015; Radford et al., 2015; Isola et al., 2016). We argue that there are two prominent limitations to such popular formalisms: (1) First and foremost, humans can imagine *multiple* plausible output images given a incomplete input. We see this rich space of potential outputs as a vital part of the human capacity to imagine and generate. Conditional GANs are in principle able to generate multiple outputs through the injection of noise, but in practice suffer from limited diversity (i.e., mode collapse) (Fig. 2). Recent approaches even remove the noise altogether, treating conditional image synthesis as regression problem (Chen & Koltun, 2017). (2) Deep networks are still difficult to explain or interpret, making the synthesized output difficult to modify. One implication is that users are not able to *control* the synthesized output. Moreover, the right mechanism for even specifying user constraints (e.g., "generate a cat image that looks like my cat") is unclear. This restricts applicability, particularly for graphics tasks.

**Nearest-neighbors:** To address these limitations, we appeal to a classic learning architecture that can naturally allow for multiple outputs and user-control: non-parametric models, or nearest-neighbors (NN). Though quite a classic approach (Efros & Leung, 1999; Efros & Freeman, 2001;

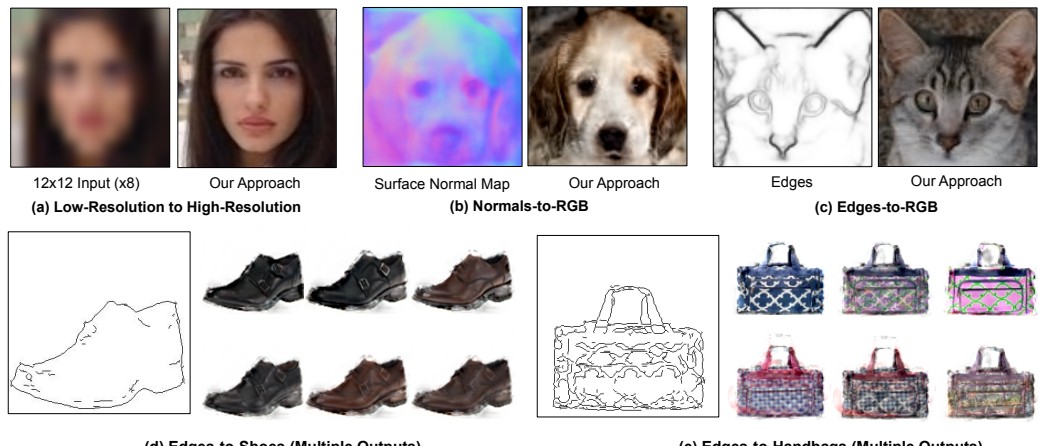

Figure 1: Our approach generates photorealistic output for various "incomplete" signals such as a low resolution image, a surface normal map, and edges/boundaries for human faces, cats, dogs, shoes, and handbags. Importantly, our approach can easily generate multiple outputs for a given input which was not possible in previous approaches (Isola et al., 2016) due to mode-collapse problem. Best viewed in electronic format.

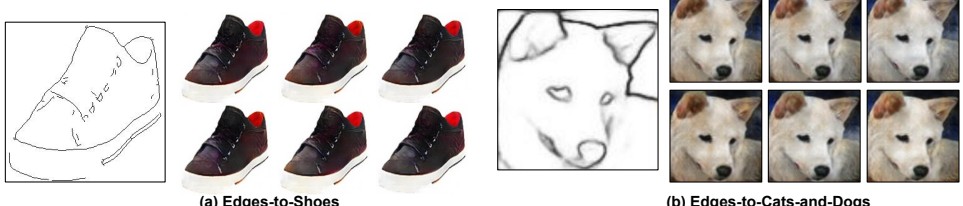

Figure 2: **Mode collapse problem for GANs:** We ran pix-to-pix pipeline of Isola et al. (2016) 72 times. Despite the random noise set using dropout at test time, we observe similar output generated each time. Here we try to show 6 possible diverse examples of generation for a hand-picked best-looking output from Isola et al. (2016).

Freeman et al., 2002; Hertzmann et al., 2001; Johnson et al., 2011), it has largely been abandoned in recent history with the advent of deep architectures. Intuitively, NN matches an incomplete input query to a large corpus of training pairs of (incomplete inputs, high-quality outputs), and simply returns the corresponding output. This trivially generalizes to multiple outputs through $K$-NN and allows for intuitive user control through on-the-fly modification of the training corpus - e.g., by restricting the training examplars to those that "look like my cat".

In practice, there are several limitations in applying NN for conditional image synthesis. The first is a practical lack of training data. The second is a lack of an obvious distance metric. And the last is a computational challenge of scaling search to large training sets.

**Approach:** To reduce the dependency on training data, we take a *compositional* approach by matching *local* pixels instead of global images. This allows us to synthesize a face by "copy-pasting" the eye of one training image, the nose of another, etc. Compositions dramatically increase the representational power of our approach: given that we want to synthesize an image of $K$ pixels using $N$ training images (with $K$ pixels each), we can synthesize an exponential number $(NK)^K$ of compositions, versus a linear number of global matches $(N)$. A significant challenge, however, is defining an appropriate feature *descriptor* for matching pixels in the incomplete input signal. We would like to capture context (such that whisker pixels are matched only to other whiskers) while allowing for compositionality (left-facing whiskers may match to right-facing whiskers). To do so, we make use of deep features, as described below.

**Pipeline:** Our precise pipeline (Figure 3) works in two stages. (1) We first train an initial regressor (CNN) that maps the incomplete input into a single output image. This output image suffers from the aforementioned limitations - it is a single output that will tend to look like a "smoothed" average

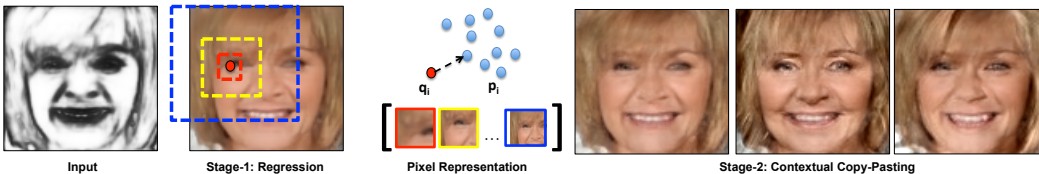

Figure 3: **Overview of pipeline:** Our approach is a two-stage pipeline. The first stage directly regresses an image from an incomplete input (using a CNN trained with $l_2$ loss). This image will tend to look like a "smoothed" average of all the potential images that could be generated. In the second stage, we look for matching pixels in similarly-smoothed training images. Importantly, we match pixels using multiscale descriptors that capture the appropriate levels of context (such that eye pixels tend to match only to eyes). To do so, we make use of off-the-shelf hypercolumn features extracted from a CNN trained for semantic pixel segmentation. By varying the size of the matched set of pixels, we can generate multiple outputs (on the right).

of all the potential images that could be generated. (2) We then perform nearest-neighbor queries on pixels *from this regressed output*. Importantly, pixels are matched (to regressed outputs from training data) using a multiscale deep descriptor that captures the appropriate level of context. This enjoys the aforementioned benefits - we can efficiently match to an exponential number of training examples in an interpretable and controllable manner. Finally, an interesting byproduct of our approach is the generation of dense, pixel-level correspondences from the training set to the final synthesized outputs.

## 2 RELATED WORK

Our work is inspired by a large body of work on discriminative and generative models, nearest neighbors architectures, pixel-level tasks, and dense pixel-level correspondences. We provide a broad overview, focusing on those most relevant to our approach.

**Synthesis with CNNs:** Convolutional Neural Networks (CNNs) have enjoyed great success for various discriminative pixel-level tasks such as segmentation (Bansal et al., 2017; Long et al., 2015), depth and surface normal estimation (Bansal et al., 2016; Eigen et al., 2013; Eigen & Fergus, 2015), semantic boundary detection (Bansal et al., 2017; Xie & Tu, 2015) etc. Such networks are usually trained using standard losses (such as softmax or $l_2$ regression) on image-label data pairs. However, such networks do not typically perform well for the *inverse* problem of image synthesis from a (in-complete) label, though exceptions do exist (Chen & Koltun, 2017). A major innovation was the introduction of adversarially-trained generative networks (GANs) (Goodfellow et al., 2014). This formulation was hugely influential in computer visions, having been applied to various image generation tasks that condition on a low-resolution image (Denton et al., 2015; Ledig et al., 2016), segmentation mask (Isola et al., 2016), surface normal map (Wang & Gupta, 2016) and other inputs (Chen et al., 2016; Huang et al., 2016; Radford et al., 2015; Wu et al., 2016; Zhang et al., 2016a; Zhu et al., 2017). Most related to us is Isola et al. (2016) who proposed a general loss function for adversarial learning, applying it to a diverse set of image synthesis tasks. Importantly, they report the problem of mode collapse, and so cannot generate diverse outputs nor control the synthesis with user-defined constraints (unlike our work).

**Interpretability and user-control:** Interpreting and explaining the outputs of generative deep networks is an open problem. As a community, we do not have a clear understanding of what, where, and how outputs are generated. Our work is fundamentally based on *copy-pasting* information via nearest neighbors, which explicitly reveals how each pixel-level output is generated (by in turn revealing where it was copied from). This makes our synthesized outputs quite interpretable. One important consequence is the ability to intuitively edit and control the process of synthesis. Zhu et al. (2016) provide a user with controls for editing image such as color, and outline. But instead of using a predefined set of editing operations, we allow a user to have an *arbitrarily*-fine level of control through on-the-fly editing of the exemplar set (e.g., "resynthesize the output using the eye from this training image and the nose from that one").

**Correspondence:** An important byproduct of pixelwise NN is the generation of pixelwise correspondences between the synthesized output and training examples. Establishing such pixel-level

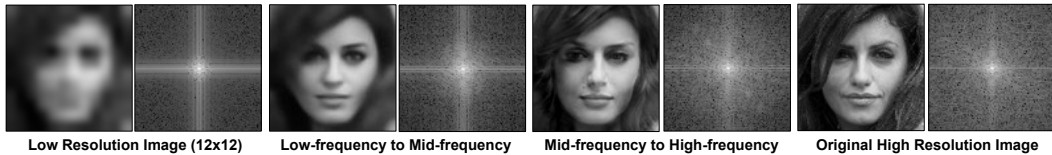

| Low Resolution Image (12x12) | Low-frequency to Mid-frequency | Mid-frequency to High-frequency | Original High Resolution Image |

Figure 4: **Frequency Analysis:** We show the image and its corresponding Fourier spectrum. Note how the frequency spectrum improve as we move from left to right. The Fourier spectrum of our final output closely matches that of original high resolution image.

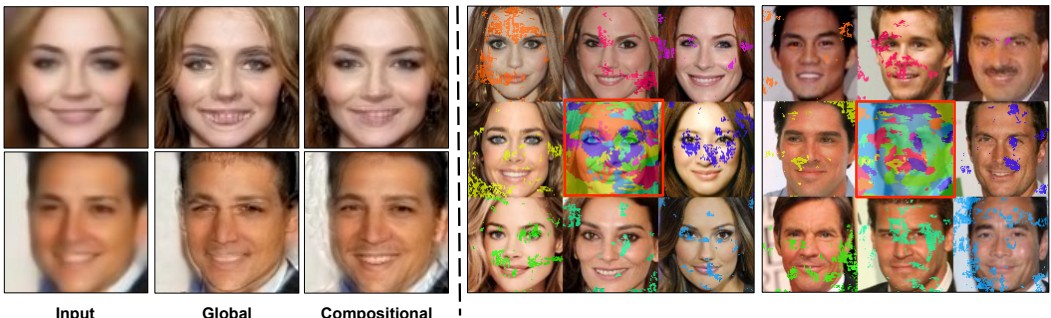

| Input | Global | Compositional |

Figure 5: **Global vs. Compositional:** Given the low-resolution input images on the left, we show high-frequency output obtained with a global nearest neighbor versus a compositional reconstruction. We visualize the correspondences associated with the compositional reconstruction on the right. We surround the reconstruction with 8 neighboring training examples, and color code pixels to denote correspondences. For example, when reconstructing the female face, forehead pixels are copied from the top-left neighbor (orange) , while right-eye pixels are copied from the bottom-left neighbor (green).

correspondence has been one of the core challenges in computer vision (Choy et al., 2016; Kanazawa et al., 2016; Liu et al., 2011; Long et al., 2014; Wei et al., 2015; Zhou et al., 2016a;b). Tappen & Liu (2012) used SIFT flow (Liu et al., 2011) to hallucinate details for image super-resolution. Zhou et al. (2016b) proposed a CNN to predict appearance flow that can be used to transfer information from input views to synthesize a new view. Kanazawa et al. (2016) generate 3D reconstructions by training a CNN to learn correspondence between object instances. Our work follows from the crucial observation of Long et al. (2014), who suggested that features from pre-trained convnets can also be used for pixel-level correspondences. In this work, we make an additional empirical observation: hypercolumn features trained for semantic segmentation learn nuances and details better than one trained for image classification. This finding helped us to establish semantic correspondences between the pixels in query and training images, and enabled us to extract high-frequency information from the training examples to synthesize a new image from a given input.

**Nonparametrics:** Our work closely follows data-driven approaches that make use of nearest neighbors (Efros & Leung, 1999; Efros & Freeman, 2001; Freeman et al., 2000; 2002; Ren et al., 2005; Hays & Efros, 2007; Johnson et al., 2011; Shrivastava et al., 2011). Hays & Efros (2007) match a query image to 2 million training images for various tasks such as image completion. We make use of dramatically smaller training sets by allowing for compositional matches. Liu et al. (2007) propose a two-step pipeline for face hallucination where global constraints capture overall structure, and local constraints produce photorealistic local features. While they focus on the task of facial super-resolution, we address variety of synthesis applications.

Final, our compositional approach is inspired by Boiman & Irani (2006; 2007), who reconstruct a query image via compositions of training examples.

## 3 PIXELNN: ONE-TO-MANY MAPPINGS

We define the problem of conditional image synthesis as follows: given an input $x$ to be conditioned on (such as an edge map, normal depth map, or low-resolution image), synthesize a high-quality output image(s). To describe our approach, we focus on illustrative the task of image super-resolution,

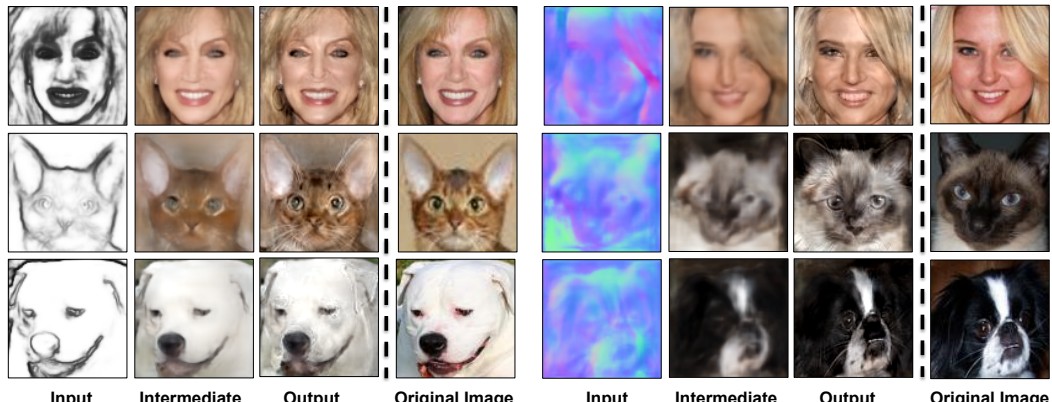

Figure 6: **Edges/Normals to RGB:** Our approach used for faces, cats, and dogs to generate RGB maps for a given edge/normal map as input. One output was picked from the multiple generations.

where the input is a low-resolution image. We assume we are given training pairs of input/outputs, written as $(x_n, y_n)$. The simplest approach would be formulating this task as a (nonlinear) regression problem:

$$\min_{w} ||w||^2 + \sum_n ||y_n - f(x_n; w)||^2 \tag{1}$$

where $f(x_n; w)$ refers to the output of an arbitrary (possibly nonlinear) regressor parameterized with $w$. In our formulation, we use a fully-convolutional neural net – specifically, PixelNet (Bansal et al., 2017) – as our nonlinear regressor. For our purposes, this regressor could be any trainable black-box mapping function. But crucially, such functions generate *one-to-one* mappings, while our underlying thesis is that conditional image synthesis should generate *many* mappings from an input. By treating synthesis as a regression problem, it is well-known that outputs tend to be over-smoothed (Johnson et al., 2016). In the context of the image colorization task (where the input is a grayscale image), such outputs tend to desaturated (Larsson et al., 2016; Zhang et al., 2016b).

**Frequency analysis:** Let us analyze this smoothing a bit further. Predicted outputs $f(x)$ (we drop the dependance on $w$ to simplify notation) are particularly straightforward to analyze in the context of super-resolution (where the conditional input $x$ is a low-resolution image). Given a low-resolution image of a face, there may exist multiple textures (e.g., wrinkles) or subtle shape cues (e.g., of local features such as noses) that could be reasonably generated as output. In practice, this set of outputs tends to be "blurred" into a single output returned by a regressor. This can be readily seen in a frequency analysis of the input, output, and original target image (Fig. 4). In general, we see that the regressor generates mid-frequencies fairly well, but fails to return much high-frequency content. We make the operational assumption that a single output suffices for mid-frequency output, but *multiple* outputs are required to capture the space of possible high-frequency textures.

**Global/Exemplar Matching:** To capture multiple possible outputs, we appeal to a classic non-parametric approaches in computer vision. We note that a simple K-nearest-neighbor (KNN) algorithm has the trivial ability to report back $K$ outputs. However, rather than using a KNN model to return an entire image, we can use it to predict the (multiple possible) high-frequencies missing from $f(x)$:

$$Global(x) = f(x) + \left( y_k - f(x_k) \right) \qquad \text{where} \qquad k = \operatorname*{argmin}_n \text{Dist}\left( f(x), f(x_n) \right) \tag{2}$$

where $Dist$ is some distance function measuring similarity between two (mid-frequency) reconstructions. To generate multiple outputs, one can report back the $K$ best matches from the training set instead of the overall best match.

**Compositional Matching:** However, the above is limited to report back high frequency images in the training set. As we previously argued, we can synthesize a much larger set of outputs by *copying* and *pasting* (high-frequency) patches from the training set. To allow for such compositional matchings, we simply match individual pixels rather than global images. Writing $f_i(x)$ for the $i^{th}$

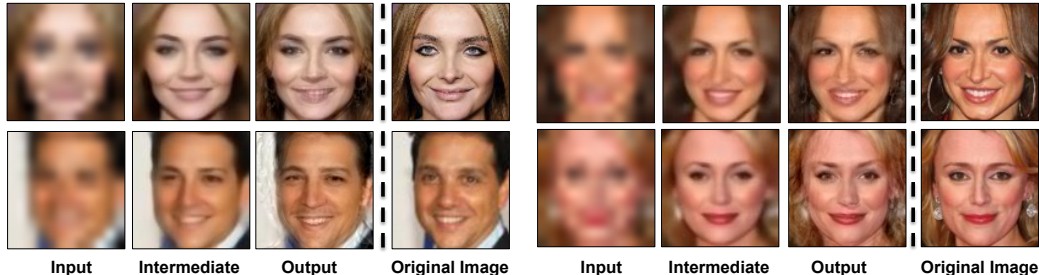

| Input | Intermediate | Output | Original Image | Input | Intermediate | Output | Original Image |

Figure 7: **Low-Resolution to High-Resolution:** We used our approach for hallucinating $96{\times}96$ images from an input $12{\times}12$ low-resolution image. One output was picked from multiple generations.

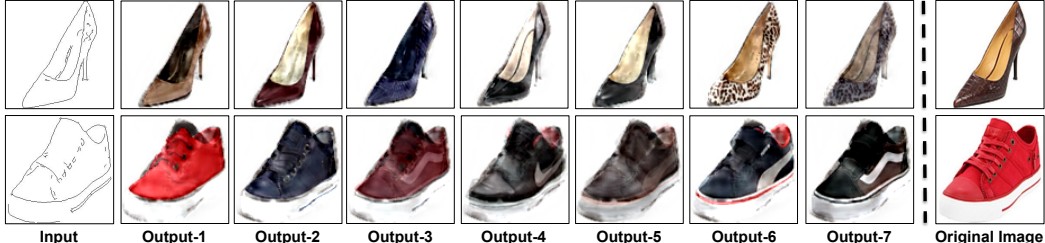

| Input | Output-1 | Output-2 | Output-3 | Output-4 | Output-5 | Output-6 | Output-7 | Original Image |

Figure 8: **Edges-to-Shoes:** Our approach used to generate multiple outputs of shoes from the edges. We picked seven distinct examples from multiple generations.

pixel in the reconstructed image, the final composed output can be written as:

$$Comp_i(x) = f_i(x) + \Big(y_{jk} - f_j(x_k)\Big) \qquad \text{where} \quad (j,k) = \underset{m,n}{\operatorname{argmin}} \operatorname{Dist}\Big(f_i(x), f_m(x_n)\Big) \quad (3)$$

where $y_{jk}$ refers to the output pixel $j$ in training example $k$.

**Distance function & Pixel representation:** A crucial question in non-parametric matching is the choice of distance function. To compare global images, contemporary approaches tend to learn a deep embedding where similarity is preserved (Bell & Bala, 2015; Chopra et al., 2005; Long et al., 2015). Distance functions for pixels are much more subtle (3). In theory, one could also learn a metric for pixel matching, but this requires large-scale training data with dense pixel-level correspondences.

Suppose we are trying to generate the left corner of an eye. If our distance function takes into account only local information around the corner, we might mistakenly match to the other eye or mouth. If our distance function takes into account only global information, then compositional matching reduces to global (exemplar) matching. Instead, we exploit the insight from previous works that different layers of a deep network tend to capture different amounts of spatial context (due to varying receptive fields) (Bansal et al., 2017; Hariharan et al., 2015; Raiko et al., 2012; Sermanet et al., 2013). Hypercolumn descriptors (Hariharan et al., 2015) aggregate such information across multiple layers into a highly accurate, *multi-scale* pixel representation (visualized in Fig. 3). We construct a pixel descriptor using features from conv-$\{1_2, 2_2, 3_3, 4_3, 5_3\}$ for a PixelNet model trained for semantic segmentation (on PASCAL Context (Mottaghi et al., 2014)).

To measure pixel similarity, we compute cosine distances between two descriptors. We visualize the compositional matches (and associated correspondences) in Figure 5. Finally, Figure 6, and Figure 7 shows the output of our approach for various input modalities.

**Efficient search:** We have so far avoided the question of run-time for our pixel-wise NN search. A naive approach would be to exhaustively search for every pixel in the dataset but that would make the computation vary linearly with the size of dataset. On the other hand, deep generative models outpace naive NN search, which is one of the reasons for their popularity over NN search.

To speed up search, we made some non-linear approximations: Given a reconstructed image $f(x)$, we first (1) find the global K-NN using *conv-5* features and then (2) search for pixel-wise matches only in a $T \times T$ pixel window around pixel $i$ in this set of $K$ images. In practice, we vary $K$

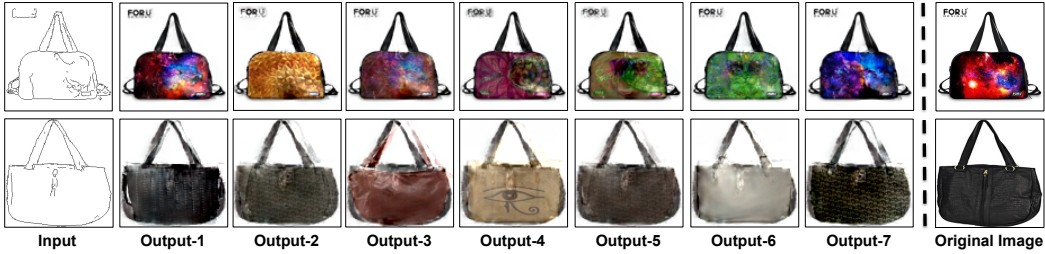

Figure 9: **Edges-to-Bags:** Our approach used to generate multiple outputs of bags from the edges. We picked seven distinct examples from multiple generations.

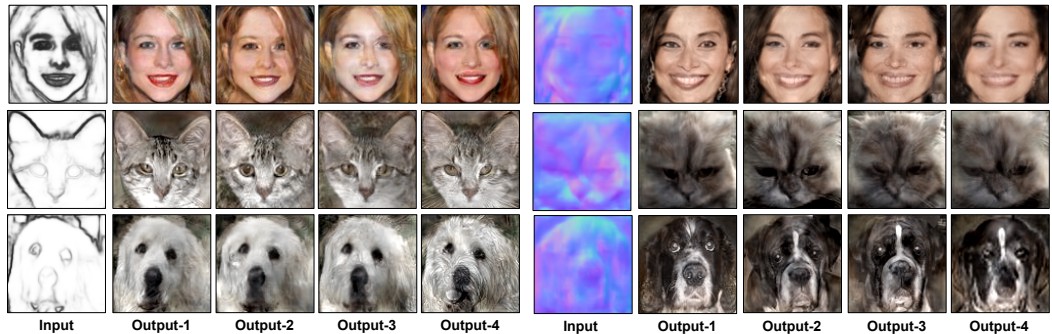

Figure 10: **Multiple Outputs for Edges/Normals to RGB:** Our approach used to generate multiple outputs of faces, cats, and dogs from the edges/normals. As an example, note how the subtle details such as eyes, stripes, and whiskers of cat (left) that could not be inferred from the edge map are different in multiple generations.

from $\{1, 2, .., 10\}$ and $T$ from $\{1, 3, 5, 10, 96\}$ and generate 72 candidate outputs for a given input. Because the size of synthesized image is 96×96, our search parameters include both a fully-compositional output ($K = 10, T = 96$) and a fully global exemplar match ($K = 1, T = 1$) as candidate outputs. Our approximate neighbor neighbor search takes .2 fps. We did not optimize our approach for speed. Importantly, we make use of a single CPU to perform our nearest neighbor search, while Isola et al. (2016)makes use of a GPU. We posit that GPU-based nearest-neighbor libraries (e.g., FAISS) will allow for real-time performance comparable to Isola et al. (2016).

Figure 8, Figure 9, and Figure 10 show examples of multiple outputs generated using our approach by simply varying these parameters.

# 4 EXPERIMENTS

We now present our findings for multiple modalities such as a low-resolution image (12×12 image), a surface normal map, and edges/boundaries for domains such as human faces, cats, dogs, handbags, and shoes. We compare our approach both quantitatively and qualitatively with the recent work of Isola et al. (2016) that use generative adversarial networks for pixel-to-pixel translation.

**Dataset:** We conduct experiments for human faces, cats and dogs, shoes, and handbags using various modalities.

**Human Faces** We use $100,000$ images from the training set of CUHK CelebA dataset (Liu et al., 2015) to train a regression model and do NN. We used the subset of test images to evaluate our approach. The images were resized to 96×96 following Gucluturk et al. (2016).

**Cats and Dogs:** We use $3,686$ images of cats and dogs from the Oxford-IIIT Pet dataset (Parkhi et al., 2012). Of these $3,000$ images were used for training, and remaining $686$ for evaluation. We used the bounding box annotation made available by Parkhi et al. (2012) to extract head of the cats and dogs.

| Normals-to-RGB | Mean | Median | RMSE | 11.25° | 22.5° | 30° | AP |
|---|---|---|---|---|---|---|---|
| **Human Faces** | | | | | | | |
| Pix-to-Pix | 17.2 | 14.3 | 21.0 | 37.2 | 74.7 | 86.8 | 0.34 |
| Pix-to-Pix (Oracle) | 15.8 | 13.1 | 19.4 | 41.9 | 78.5 | 89.3 | 0.34 |
| PixelNN (Rand-1) | 12.8 | 10.4 | 16.0 | 54.2 | 86.6 | 94.1 | 0.38 |
| PixelNN (Oracle) | **10.8** | **8.7** | **13.5** | **63.7** | **91.6** | **96.7** | **0.42** |
| **Cats and Dogs** | | | | | | | |
| Pix-to-Pix | 14.7 | 12.8 | 17.5 | 42.6 | 82.5 | 92.9 | 0.82 |
| Pix-to-Pix (Oracle) | **13.2** | **11.4** | **15.7** | **49.2** | **87.1** | **95.3** | 0.85 |
| PixelNN (Rand-1) | 16.6 | 14.3 | 19.8 | 36.8 | 76.2 | 88.8 | 0.80 |
| PixelNN (Oracle) | 13.8 | 11.9 | 16.6 | 46.9 | 84.9 | 94.1 | **0.92** |

| Edges-to-RGB | AP | Mean | Median | RMSE | 11.25° | 22.5° | 30° |
|---|---|---|---|---|---|---|---|
| **Human Faces** | | | | | | | |
| Pix-to-Pix | 0.35 | 12.1 | 9.6 | 15.5 | 58.1 | 88.1 | 94.7 |
| Pix-to-Pix(Oracle) | 0.35 | 11.5 | 9.1 | 14.6 | 61.1 | 89.7 | 95.6 |
| PixelNN (Rand-1) | 0.38 | 13.3 | 10.6 | 16.8 | 52.9 | 85.0 | 92.9 |
| PixelNN (Oracle) | **0.41** | **11.3** | **9.0** | **14.4** | **61.6** | **90.0** | **95.7** |
| **Cats and Dogs** | | | | | | | |
| Pix-to-Pix | 0.78 | 18.2 | 16.0 | 21.8 | 32.4 | 71.0 | 85.1 |
| Pix-to-Pix (Oracle) | 0.81 | 16.5 | 14.2 | 19.8 | 37.2 | 76.4 | 89.0 |
| PixelNN (Rand-1) | 0.77 | 18.9 | 16.4 | 22.5 | 30.3 | 68.9 | 83.5 |
| PixelNN (Oracle) | **0.89** | **16.3** | **14.1** | **19.6** | **37.6** | **77.0** | **89.4** |

Table 1: We compared our approach, PixelNN, with the GAN-based formulation of Isola et al. (2016) for human faces, and cats and dogs. We used an off-the-shelf PixelNet model trained for surface normal estimation and edge detection. We use the output from real images as ground truth surface normal and edge map respectively.

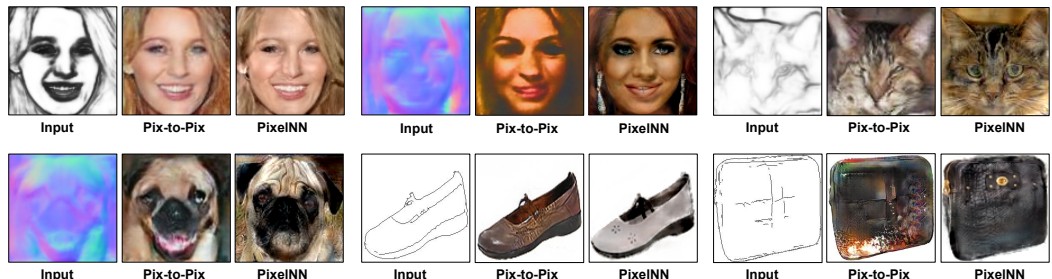

Figure 11: Comparison of our approach with Pix-to-Pix (Isola et al., 2016).

For human faces, and cats and dogs, we used the pre-trained PixelNet (Bansal et al., 2017) to extract surface normal and edge maps. We did not do any post-processing (NMS) to the outputs of edge detection.

**Shoes & Handbags:** We followed Isola et al. (2016) for this setting. $50,000$ training images of shoes were used from (Yu & Grauman, 2014), and $137,000$ images of Amazon handbags from (Zhu et al., 2016). The edge maps for this data was computed using HED (Xie & Tu, 2015) by Isola et al. (2016).

**Qualitative Evaluation:** Figure 11 shows the comparison of our NN based approach (**PixelNN**) with Isola et al. (2016) (**Pix-to-Pix**).

**Quantitative Evaluation:** We quantitatively evaluate our approach to measure if our generated outputs for human faces, cats and dogs can be used to determine surface normal and edges from an off-the-shelf trained PixelNet (Bansal et al., 2017) model for surface normal estimation and edge detection. The outputs from the real images are considered as ground truth for evaluation as it gives an indication of how far are we from them. Somewhat similar approach is used by Isola et al. (2016) to measure their synthesized cityscape outputs and compare against the output using real world images, and Wang & Gupta (2016) for object detection evaluation.

We compute six statistics, previously used by (Bansal et al., 2016; Eigen & Fergus, 2015; Fouhey et al., 2013; Wang et al., 2015), over the angular error between the normals from a synthesized image and normals from real image to evaluate the performance – **Mean**, **Median**, **RMSE**, **11.25°**, **22.5°**, and **30°** – The first three criteria capture the mean, median, and RMSE of angular error, where lower is better. The last three criteria capture the percentage of pixels within a given angular error, where higher is better. We evaluate the edge detection performance using average precision (**AP**).

Table 1 quantitatively shows the performance of our approach with (Isola et al., 2016). Our approach generates multiple outputs and we do not have any direct way of ranking the outputs, therefore we show the performance using a random selection from one of 72 outputs, and an oracle selecting the best output. To do a fair comparison, we ran trained models for Pix-to-Pix (Isola et al., 2016) 72 times and used an oracle for selecting the best output as well. We observe that our approach generates better multiple outputs as performance improves significantly from a random selection to

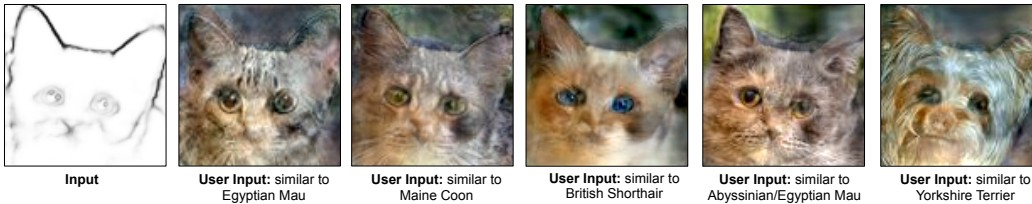

Figure 12: **Controllable synthesis:** We generate the output of cats given a user input from a edge map. From the edge map, we do not know what type of cat it is. A user can suggest what kind of the output they would like, and our approach can copy-paste the information.

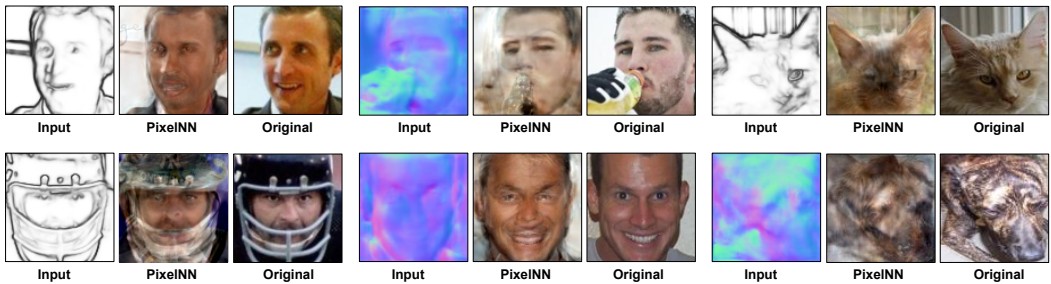

Figure 13: **Failure Cases:** We show some failure cases for different input types. Our approach mostly fails when it is not able to find suitable nearest neighbors.

oracle as compared with Isola et al. (2016). Our approach, though based on simple NN, achieves result quantitatively and qualitatively competitive (and many times better than) with state-of-the-art models based on GANs and produce outputs close to natural images.

**Controllable synthesis:** Finally, NN provides a user with intuitive control over the synthesis process. We explore a simple approach based on *on-the-fly pruning* of the training set. Instead of matching to the entire training library, a user can specify a subset of relevant training examples. Figure 12 shows an example of controllable synthesis. A user "instructs" the system to generate an image that looks like a particular dog-breed by either denoting the subset of training examplars (e.g., through a subcategory label), or providing an image that can be used to construct an on-the-fly neighbor set.

**Failure cases:** Our approach mostly fails when there are no suitable NNs to extract the information from. Figure 13 shows some example failure cases of our approach. One way to deal with this problem is to do exhaustive pixel-wise NN search but that would increase the run-time to generate the output. We believe that system-level optimization such as Scanner[1], may potentially be useful in improving the run-time performance for pixel-wise NNs.

# 5 DISCUSSION

We present a simple approach to image synthesis based on compositional nearest-neighbors. Our approach somewhat suggests that GANs themselves may operate in a compositional "copy-and-paste" fashion. Indeed, examining the impressive outputs of recent synthesis methods suggests that some amount of local memorization is happening. However, by making this process explicit, our system is able to naturally generate multiple outputs, while being interpretable and amenable to user constraints. An interesting byproduct of our approach is dense pixel-level correspondences. If training images are augmented with semantic label masks, these labels can be transfered using our correspondences, implying that our approach may also be useful for image analysis through label transfer (Liu et al., 2011).

---

[1]https://github.com/scanner-research/scanner

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
