# OpenReview forum: "PixelNN: Example-based Image Synthesis"
_ICLR.cc/2018/Conference — Accept (Poster)_

### Official Review · AnonReviewer1 · 2017-11-27
**Nice approach on conditional image generation**

**Rating:** 8
**Confidence:** 4

**Review:**

Overall I like the paper and the results look nice in a diverse set of datasets and tasks such as edge-to-image, super-resolution, etc. Unlike the generative distribution sampling of GANs, the method provides an interesting compositional scheme, where the low frequencies are regressed and the high frequencies are obtained by "copying" patches from the training set. In some cases the results are similar to pix-to-pix (also in the numerical evaluation) but the method allows for one-to-many image generation, which is a important contribution. Another positive aspect of the paper is that the synthesis results can be analyzed, providing insights for the generation process.

While most of the paper is well written, some parts are difficult to parse. For example, the introduction has some parts that look more like related work (that is mostly a personal preference in writting). Also in Section 3, the paragraph for distance functions do not provide any insight about what is used, but it is included in the next paragraph (I would suggest either merging or not highlighting the paragraphs).

Q: The spatial grouping that is happening in the compositional stage, is it solely due to the multi-scale hypercolumns?  Would the result be more inconsistent if the hypercolumns had smaller receptive field?

Q: For the multiple outputs, the k neighbor is selected at random?

---

> ### Author Response · Authors · 2017-12-16
> **Thanks for the positive feedback and suggestion to improve writing**
>
> We thank the reviewer for the suggestion to improve the writing, and will incorporate these suggestions in our final version.
>
> 1. "The spatial grouping that is happening in the compositional stage, is it solely due to the multi-scale hypercolumns?  Would the result be more inconsistent if the hypercolumns had smaller receptive field?"
>
> Yes, we think so. We believe that much of the spatial grouping is due to the multi-scale hypercolumns. The results degrade with smaller receptive fields.
>
> 2. "For the multiple outputs, the k neighbor is selected at random?"
>
> Yes, the k-neighbors are selected at random as described in "Efficient Search" on page-6. We will clarify this.

---

### Official Review · AnonReviewer3 · 2017-11-28
**Shines Light on Deficiencies in Conditional GAN: borderline accept**

**Rating:** 6
**Confidence:** 4

**Review:**

This paper presents a pixel-matching based approach to synthesizing RGB images from input edge or normal maps. The approach is compared to Isola et al’s conditional adversarial networks, and unlike the conditional GAN, is able to produce a diverse set of outputs.

Overall, the paper describes a computer visions system based on synthesizing images, and not necessarily a new theoretical framework to compete with GANs. With the current focus of the paper being the proposed system, it is interesting to the computer vision community. However, if one views the paper in a different light, namely showing some “blind-spots” of current conditional GAN approaches like lack of diversity, then it can be of much more interest to the broader ICLR community.

Pros:
Overall the paper is well-written
Makes a strong case that random noise injection inside conditional GANs does not produce enough diversity
Shows a number of qualitative and quantitative results

Concerns about the paper:
1.) It is not clear how well the proposed approach works with CNN architectures other than PixelNet
2.) Since the paper used “the pre-trained PixelNet to extract surface normal and edge maps” for ground-truth generation, it is not clear whether the approach will work as well when the input is a ground-truth semantic segmentation map.
3.) Since the paper describes a computer-vision image synthesis system and not a new theoretical result, I believe reporting the actual run-time of the system will make the paper stronger. Can PixelNN run in real-time? How does the timing compare to Isola et al’s Conditional GAN?

Minor comments:
1.) The paper mentions making predictions from “incomplete” input several times, but in all experiments, the input is an edge map, normal map, or low-resolution image. When reading the manuscript the first time, I was expecting experiments on images that have regions that are visible and regions that are masked out. However, I am not sure if the confusion is solely mine, or shared with other readers.

2.) Equation 1 contains the norm operator twice, and the first norm has no subscript, while the second one has an l_2 subscript. I would expect the notation style to be consistent within a single equation (i.e., use ||w||_2^2, ||w||^2, or ||w||_{l_2}^2)

3.) Table 1 has two sub-tables: left and right. The sub-tables have the AP column in different places.

4.) “Dense pixel-level correspondences” are discussed but not evaluated.

---

> ### Author Response · Authors · 2017-12-16
> **Thanks for the insightful comments and positive feedback**
>
> We thank the reviewer for their comments and suggestions, and appreciate their effort to highlight our work for a broader ICLR community. We will incorporate the suggestions provided in the reviews.
>
> 1. "It is not clear how well the proposed approach works with CNN architectures other than PixelNet"
>
> We will add experiments with other architectures. However, we believe that our approach is agnostic of a pixel-level CNN used for regression. We used PixelNet because it had been shown to work well for the various pixel-level tasks, particularly the inverse of our synthesis problems (i.e., predicting surface normals and edges from images). The use of a single network architecture for our various synthesis problems reduces variability due to the regressor and lets us focus on the nearest neighbor stage.
>
> 2. "Since the paper used “the pre-trained PixelNet to extract surface normal and edge maps” for ground-truth generation, it is not clear whether the approach will work as well when the input is a ground-truth semantic segmentation map.
>
> This is an interesting question. We have initial results that synthesize faces from the Helen Face dataset (Smith et al, CVPR 2013) from ground-truth segmentation masks. We see qualitatively similar behaviour. In many cases we even see better performance because the input signal (i.e., the ground-truth segmentation labels) are of higher quality than the edges/normals we condition on. We will add such an analysis and discussion.
>
> 3. "Since the paper describes a computer-vision image synthesis system and not a new theoretical result, I believe reporting the actual run-time of the system will make the paper stronger. Can PixelNN run in real-time? How does the timing compare to Isola et al’s Conditional GAN?"
>
> Our approximate neighbor neighbor search (described on Page 6) takes .2 fps. We did not optimize our approach for speed. Importantly, we make use of a single CPU to perform our nearest neighbor search, while Isola et al makes use of a GPU. We posit that GPU-based nearest-neighbor libraries (e.g., FAISS) will allow for real-time performance comparable to Isola’s. We will add a discussion.

---

### Official Review · AnonReviewer2 · 2017-12-02
**Simple and effective baseline for conditional image generation**

**Rating:** 7
**Confidence:** 3

**Review:**

This paper proposes a compositional nearest-neighbors approach to image synthesis, including results on several conditional image generation datasets.

Pros:
- Simple approach based on nearest-neighbors, likely easier to train compared to GANs.
- Scales to high-resolution images.

Cons:
- Requires a potentially costly search procedure to generate images.
- Seems to require relevant objects and textures to be present in the training set in order to succeed at any given conditional image generation task.

---

> ### Author Response · Authors · 2017-12-16
> **Thanks for the positive feedback**
>
> We thank the reviewer for their feedback.
>
> 1. "Requires a potentially costly search procedure to generate images." -
>
> We agree that this approach could be computationally expensive in its naive form. However, the use of optimized libraries such as FAISS, FLAWN etc. can be used to reduce the run-time. Similar to CNNs, the use of parallel processing modules such as GPUs could drastically reduce the time spent on search procedure.
>
>
> 2. "Seems to require relevant objects and textures to be present in the training set in order to succeed at any given conditional image generation task."
>
> We agree. However, this criticism could also be applied to most learning-based models (including CNNs and GANs, as R3 points out).

---

### Decision · Program_Chairs · 2018-01-29
**ICLR 2018 Conference Acceptance Decision**

**Decision:**

Accept (Poster)

**Comment:**

The paper proposes a novel method for conditional image generation which is based on nearest neighbor matching for transferring high-frequency statistics. The evaluation is carried out on several image synthesis tasks, where the technique is shown to perform better than an adversarial baseline.